# Offline Decentralized Multi-Agent Reinforcement Learning

## Abstract

In many real-world multi-agent cooperative tasks, due to high cost and risk, agents cannot continuously interact with the environment and collect experiences during learning, but have to learn from offline datasets. However, the transition probabilities calculated from the dataset can be much different from the transition probabilities induced by the learned policies of other agents, creating large errors in value estimates. Moreover, the experience distributions of agents' datasets may vary wildly due to diverse behavior policies, causing large difference in value estimates between agents. Consequently, agents will learn uncoordinated suboptimal policies. In this paper, we propose MABCQ, which exploits *value deviation* and *transition normalization* to modify the transition probabilities. Value deviation optimistically increases the transition probabilities of high-value next states, and transition normalization normalizes the biased transition probabilities of next states. They together encourage agents to discover potential optimal and coordinated policies. Mathematically, we prove the convergence of Q-learning under the non-stationary transition probabilities after modification. Empirically, we show that MABCQ greatly outperforms baselines and reduces the difference in value estimates between agents.

## 1 Introduction

In multi-agent cooperative tasks, agents learn from the experiences generated by continuously interacting with the environment to maximize the cumulative shared reward. Recently, multi-agent reinforcement learning (MARL) has been applied to real-world cooperative systems (Bhalla et al., 2020; Xu et al., 2021). However, in many industrial applications, continuously interacting with the environment and collecting the experiences during learning is costly, risky, and time-consuming. One way to address this is offline RL, where the agent can only access a fixed dataset of experiences and learn the policy without further interacting with the environment. However, in multi-agent environments, the dataset of each agent is often pre-collected individually by different behavior policies, which are not necessary to be expert, and each dataset contains the individual action of the agent instead of the joint action of all agents, *e.g.*, autonomous driving dataset. Therefore, the dataset does not satisfy the paradigm of centralized training, and the agent has to learn the coordinated policy in an offline and fully decentralized way.

The main challenge of offline RL is the *extrapolation error*, an error in value estimate incurred by the mismatch between the experience distributions of the learned policy and the dataset (Fujimoto et al., 2019), *e.g.*, the distance of the learned action distribution to the behavior action distribution, and the bias of the transition dynamics estimated from the dataset to the true transition dynamics. Recently, almost all offline RL methods (Fujimoto et al., 2019; Levine et al., 2020; Jaques et al., 2019) focus on constraining the learned policy to be close to the behavior policy to avoid the overestimate of the values of out-of-distribution actions, but ignore to correct the transition bias since the deviation of estimated transition dynamics would not be too large if the single-agent environment is stationary.

However, in decentralized multi-agent environments, from the perspective of each agent, other agents are a part of the environment, and the transition dynamics experienced by each agent depend on the policies of other agents. Even in a stationary environment, the experienced transition dynamics of each agent will change as other agents update their policies (Foerster et al., 2017). Since the behavior policies of other agents would be inconsistent with their learned policies which

are unknowable in decentralized multi-agent environments, the transition dynamics estimated from the dataset by each agent would be different from the transition dynamics induced by the learned policies of other agents, causing large errors in value estimates. The extrapolation error would lead to suboptimal policies. Moreover, trained on different distributions of experiences collected by various behavior policies, the estimated values of the same state might be much different between agents, which causes that the learned policies cannot coordinate with each other.

To overcome the suboptimum and miscoordination caused by transition bias in decentralized learning, we introduce *value deviation* and *transition normalization* to deliberately modify the estimated transition probabilities from the dataset. During data collection, if one agent takes an optimal action while other agents take suboptimal actions at a state, the transition probabilities of low-value next states will become large. Thus, the Q-value of the optimal action will be underestimated, and the agents will fall into suboptimum. Since the other agents are also trained, the learned policies of other agents would become better than the behavior policies. For each agent, the transition probabilities of high-value next states induced by the learned policies would be larger than those estimated from the dataset. Therefore, we let each agent be optimistic toward other agents and multiply the transition probabilities by the deviation of the value of next state from the expected value over all next states, to make the estimated transition probabilities close to the transition probabilities induced by the learned policies of other agents.

Value deviation could decrease the extrapolation error and help the agents escape from the suboptimum. However, in some cases, the behavior policies of other agents might be highly deterministic, which makes the distribution of experiences unbalanced. If the transition probabilities of high-value next states are extremely low, value deviation may not remedy the underestimate. Moreover, due to the diversity in experience distributions of agents, the value of the same state might be overestimated by some agents while underestimated by others, which results in miscoordination of the learned policies. To address these two problems, we normalize the transition probability estimated from the dataset to be uniform. Transition normalization balances the extremely biased distribution of experiences and builds the consensus about value estimate. By combining value deviation and transition normalization, the agents would learn high-performing and coordinated policies in an offline and fully decentralized way.

Although value deviation and transition normalization make transition dynamics non-stationary, we mathematically prove the convergence of Q-learning under such non-stationary transition dynamics. By importance sampling, value deviation and transition normalization take effect only as the weights of the objective function, which makes our method easy to implement. We instantiate the proposed method on BCQ, termed MABCQ, to additionally avoid out-of-distribution actions. Nevertheless, it can also be implemented on other offline RL methods. We build the offline datasets and evaluate MABCQ in four multi-agent mujoco scenarios (Todorov et al., 2012). Experimental results show that MABCQ greatly outperforms BCQ, and ablation studies demonstrate the effectiveness of value deviation and transition normalization. To the best of our knowledge, MABCQ is the *first* method for offline and fully decentralized multi-agent reinforcement learning.

## 2  RELATED WORK

**MARL.** Many MARL methods have been proposed for learning to solve cooperative tasks in an online manner. Some methods (Lowe et al., 2017; Foerster et al., 2018; Iqbal & Sha, 2019) extend policy gradient into multi-agent cases. Value factorization methods (Sunehag et al., 2018; Rashid et al., 2018; Son et al., 2019) decompose the joint value function into individual value functions. Communication methods (Das et al., 2019; Ding et al., 2020) share information between agents for better cooperation. All these methods follow centralized learning and decentralized execution, where the agents could access the information from other agents during centralized training. However, in our offline and decentralized setting, the datasets of agents are different; each dataset contains individual actions instead of joint actions; and the agents cannot be trained in a centralized way.

For decentralized learning, the key challenge is the obsolete experiences in replay buffer. Fingerprints (Foerster et al., 2017) deals with obsolete experience problems by conditioning the value function on a fingerprint that disambiguates the age of the sampled data. Lenient-DQN (Palmer et al., 2018) extends the leniency concept and introduces optimism in the value function update by forgiving suboptimal actions. Concurrent experience replay (Omidshafiei et al., 2017) induces cor-

relations in local policy updates, making agents tend to converge to the same equilibrium. However, these methods require additional information, *e.g.*, training iteration number, exploration rate, and timestamp, which often are not provided by the offline dataset.

**Offline RL.** Offline RL requires the agent to learn from a fixed batch of data $\{(s, a, s', r)\}$, consisting of single-step transitions without exploration. Unlike imitation learning, offline RL does not assume that the offline data is provided by a high-performing expert but has to handle the data generated by suboptimal or multi-modal behavior policies. Most offline RL methods consider the out-of-distribution action (Levine et al., 2020) as the fundamental challenge, which is the main cause of the extrapolation error (Fujimoto et al., 2019) in value estimate in the single-agent environment. To minimize the extrapolation error, some recent methods introduce constraints to enforce the learned policy to be close to the behavior policy, which could be direct action constraint (Fujimoto et al., 2019), kernel MMD (Kumar et al., 2019), Wasserstein distance (Wu et al., 2019), and KL divergence (Peng et al., 2019). Some methods train a Q-function pessimistic to out-of-distribution actions to avoid overestimation by adding a reward penalty quantified by the learned environment model (Yu et al., 2020), by minimizing the Q-values of out-of-distribution actions (Kumar et al., 2020; Yu et al., 2021), or by weighting the update of Q-function via Monte Carlo dropout (Wu et al., 2021). A finite-sample analysis for offline MARL (Zhang et al., 2018) has been studied, but the agents are assumed to get individual rewards instead of the shared reward, and be connected by communication networks, which is not the fully decentralized setting. All these methods do not consider the extrapolation error introduced by the transition bias, which is a fatal problem in offline and decentralized MARL.

## 3 METHOD

### 3.1 PRELIMINARIES

We consider $N$ agents in multi-agent MDP (Oliehoek & Amato, 2016) $M_{env} =< \mathcal{S}, \mathcal{A}, R, P_{env}, \gamma >$ with the state space $\mathcal{S}$ and the joint action space $\mathcal{A}$. At each timestep, each agent $i$ gets state $s$ and performs an individual action $a_i$, and the environment transitions to the next state $s'$ by taking the joint action $\vec{a}$ with the transition probability $P_{env}(s'|s, \vec{a})$. The agents would get a shared reward $r = R(s)$, which is simplified to just depending on state (Schulman et al., 2015). The agents learn to maximize the expected return $\mathbb{E} \sum_{t=0}^{T} \gamma^t r_t$, where $\gamma$ is a discount factor and $T$ is the time horizon of the episode. However, in the fully decentralized learning, $M_{env}$ is partially observable to the agent since the agent cannot observe the joint action $\vec{a}$. During execution, from the perspective of each agent $i$, there is a viewed MDP $M_{E_i} =< \mathcal{S}, \mathcal{A}_i, R, P_{E_i}, \gamma >$ with the individual action space $\mathcal{A}_i$ and the transition probability

$$P_{E_i}(s'|s, a_i) = \sum_{\vec{a}_{-i}} P_{env}(s'|s, \vec{a}) \prod_{j \neq i}^{N} \pi_j(a_j|s),$$

where $\vec{a}_{-i}$ denotes the joint action of all agents except agent $i$, and $\pi$ denotes the policy of the agent. As the transition probability depends on the policies of other agents, if other agents are also updating their policies, $P_{E_i}$ becomes non-stationary. Moreover, if the agent cannot interact with the environment, $P_{E_i}$ is unknown. Since we only investigate the influence of other agents' policies on $P_{E_i}$, we assume $P_{env}$ to be deterministic.

In *offline and decentralized settings*, each agent $i$ could only access a fixed offline dataset $\mathcal{B}_i$, which is pre-collected by behavior policies and contains the tuples $(s, a_i, r, s')$. As defined in BCQ (Fujimoto et al., 2019), the visible MDP $M_{\mathcal{B}_i} =< \mathcal{S}, \mathcal{A}_i, R, P_{\mathcal{B}_i}, \gamma >$ is constructed on $\mathcal{B}_i$, which has the transition probability[1]

$$P_{\mathcal{B}_i}(s'|s, a_i) = \frac{\text{num}(s, a_i, s')}{\sum_{\hat{s}'} \text{num}(s, a_i, \hat{s}')},$$

where $\text{num}(s, a_i, s')$ is the number of times the tuple $(s, a_i, s')$ is observed in $\mathcal{B}_i$. However, since the learned policies of other agents might be greatly different from the behavior policies, $P_{\mathcal{B}_i}$ would be biased from $P_{E_i}$, which creates large extrapolation errors and differences in value estimates between agents, and eventually leads to uncoordinated suboptimal policies.

---

[1]The transition probability in the following sections means the one calculated from $\mathcal{B}_i$ unless otherwise stated.

Table 1: The matrix game.

|  |  | Agent 2 | |
|---|---|---|---|
|  |  | $a_1$ (0.4) | $a_2$ (0.6) |
| Agent 1 | $a_1$ (0.8) | 1 | 5 |
|  | $a_2$ (0.2) | 6 | 1 |

Table 2: Transition probabilities and expected returns calculated in the dataset.

| | action | transition | expected return | | action | transition | expected return |
|---|---|---|---|---|---|---|---|
| Agent 1 | $a_1$ | $p(1\|a_1) = 0.4$
$p(5\|a_1) = 0.6$ | 3.4 | Agent 2 | $a_1$ | $p(1\|a_1) = 0.8$
$p(6\|a_1) = 0.2$ | 2.0 |
| | $a_2$ | $p(6\|a_2) = 0.4$
$p(1\|a_2) = 0.6$ | 3.0 | | $a_2$ | $p(5\|a_2) = 0.8$
$p(1\|a_2) = 0.2$ | 4.2 |

To intuitively illustrate the miscoordination caused by the transition bias, we devise offline datasets in a matrix game for two agents, with the payoff depicted in Table 1. The action distributions of the behavior polices of the two agents are $[0.8, 0.2]$ and $[0.4, 0.6]$, respectively. Table 2 shows the transition probabilities and expected returns calculated by the independent agents in the datasets. Since the datasets are collected by poor behavior policies, when one agent chooses the optimal action, the other agent would choose the suboptimal action with a high probability, which leads to low transition probabilities of high-value next states. Thus, the agents underestimate the optimal actions and converge to the suboptimal policies $(a_1, a_2)$, rather than the optimal policies $(a_2, a_1)$.

## 3.2 IMPORTANCE WEIGHTS

### 3.2.1 VALUE DEVIATION

If the behavior policies of some agents are low-performing during data collection, they usually take suboptimal actions to cooperate with the optimal actions of other agents, which leads to high transition probabilities of low-value next states. When agent $i$ performs Q-learning with the dataset $\mathcal{B}_i$, the Bellman operator $\mathcal{T}$ is approximated by the transition probability $P_{\mathcal{B}_i}(s'|s, a_i)$ to estimate the expectation over $s'$:

$$\mathcal{T}Q_i(s, a_i) = \mathbb{E}_{s' \sim P_{\mathcal{B}_i}(s'|s,a_i)} \left[ r + \gamma \max_{\hat{a}_i} Q_i(s', \hat{a}_i) \right].$$

If $P_{\mathcal{B}_i}$ of a high-value $s'$ is lower than $P_{E_i}$, the Q-value of this $(s, a_i)$ pair is underestimated, which would cause large extrapolation error and guide the agent to the convergence of suboptimal policy.

As discussed before, during execution, the transition probability viewed by agent $i$ is $P_{E_i}$ and the environment is deterministic, thus $P_{E_i}(s'|s, a_i)$ only depends on the learned policies of other agents, which are unavailable. However, since the policies of other agents are also updating towards maximizing the Q-values, $P_{E_i}$ of high-value next states would grow higher than $P_{\mathcal{B}_i}$. Based on this intuition, we let each agent be optimistic towards other agents and modify $P_{\mathcal{B}_i}$ as

$$P_{\mathcal{B}_i}(s'|s, a_i) * (1 + \underbrace{\frac{V_i^*(s') - \mathbb{E}_{\hat{s}'} V_i^*(\hat{s}')}{|\mathbb{E}_{\hat{s}'} V_i^*(\hat{s}')|}}_{\text{value deviation}}) * \frac{1}{z_i^{vd}},$$

where the state value $V_i^*(s) = \max_{a_i} Q_i(s, a_i)$, $1 + \frac{V_i^*(s') - \mathbb{E}_{\hat{s}'} V_i^*(\hat{s}')}{|\mathbb{E}_{\hat{s}'} V_i^*(\hat{s}')|}$ is the deviation of the value of next state from the expected value over all next states, which increases the transition probabilities of the high-value next states and decreases those of the low-value next states, and $z_i^{vd} = \sum_{s'} P_{\mathcal{B}_i}(s'|s, a_i) * (1 + \frac{V_i^*(s') - \mathbb{E}_{\hat{s}'} V_i^*(\hat{s}')}{|\mathbb{E}_{\hat{s}'} V_i^*(\hat{s}')|})$ is a normalization term to make sure the sum of the transition probabilities is one. *Value deviation* makes the transition probability to be close to $P_{E_i}$ and hence decreases the extrapolation error. The optimism towards other agents helps the agents escape from local optima and discover potential optimal actions which are hidden by the poor behavior policies.

### 3.2.2 TRANSITION NORMALIZATION

In real-world applications, the action distribution of behavior policy might be unbalanced, which makes the transition probabilities $P_{\mathcal{B}_i}$ biased, *e.g.*, the transition probabilities of Agent 2 (*i.e.*, action distribution of Agent 1) in Table 1. If the transition probability of a high-value next state is extremely low, *value deviation* cannot correct the underestimate. Moreover, since $\mathcal{B}_i$ of each agent is individually collected by different behavior policies, the diversity in transition probabilities of agents leads to that the value of the same state $s$ will be overestimated by some agents, while be underestimated by others. Since the agents are trained to reach high-value states, the large divergences on state values will cause miscoordination of the learned policies. To overcome these problems, we normalize the biased transition probability $P_{\mathcal{B}_i}$ to be uniform over next states,

$$P_{\mathcal{B}_i}\left(s'|s, a_i\right) * \underbrace{\frac{1}{P_{\mathcal{B}_i}\left(s'|s, a_i\right)}}_{\text{transition normalization}} * \frac{1}{z_i^{tn}},$$

where $z_i^{tn}$ is a normalization term that is the number of different $s'$ given $(s, a_i)$ in $\mathcal{B}_i$. *Transition normalization* enforces that each agent has the same $P_{\mathcal{B}_i}$ when it acts the learned action $a_i^*$ on the same state $s$, and we have the following proposition.

**Proposition 1.** *In episodic environments, if each agent $i$ performs Q-learning on $\mathcal{B}_i$, all agents will converge to the same $V^*$ if they have the same transition probability on any state where each agent $i$ acts the learned action $a_i^*$.*

*Proof.* The proof is provided in Appendix A. □

However, to satisfy $P_{\mathcal{B}_1}\left(s'|s, a_1^*\right) = P_{\mathcal{B}_2}\left(s'|s, a_2^*\right) = \ldots = P_{\mathcal{B}_N}\left(s'|s, a_N^*\right)$ for all $s' \in \mathcal{S}$, the agents should have the same set of $s'$ at $(s, a^*)$, which is a strong assumption. In practice, although the assumption is not strictly satisfied, *transition normalization* could still normalize the biased transition distribution, encouraging the estimated state value $V^*$ to be close to each other.

### 3.2.3 OPTIMIZATION OBJECTIVE

We combine *value deviation* $(1 + \frac{V_i^*(s') - \mathbb{E}_{\hat{s}'} V_i^*(\hat{s}')}{|\mathbb{E}_{\hat{s}'} V_i^*(\hat{s}')|})$, which is denoted as $\lambda_{vd_i}$, and *transition normalization* $\frac{1}{P_{\mathcal{B}_i}(s'|s, a_i)}$, which is denoted as $\lambda_{tn_i}$, and modify $P_{\mathcal{B}_i}$ as,

$$\hat{P}_{\mathcal{B}_i}\left(s'|s, a_i\right) = P_{\mathcal{B}_i}\left(s'|s, a_i\right) * \frac{\lambda_{tn_i} \lambda_{vd_i}}{z_i},$$

where $z_i = \sum_{s'}(1 + \frac{V_i^*(s') - \mathbb{E}_{\hat{s}'} V_i^*(\hat{s}')}{|\mathbb{E}_{\hat{s}'} V_i^*(\hat{s}')|})$ is the normalization term. In a sense, $\hat{P}_{\mathcal{B}_i}$ makes the offline learning on $\mathcal{B}_i$ similar to the *online* decentralized MARL. In the initial stage, $\lambda_{vd_i}$ is close to 1 since $Q_i(s, a_i)$ is not updated, and the transition probabilities are uniform, meaning other agents are acting randomly. During training, the transition probabilities of high-value states gradually grow under *value deviation*, which is an analogy of that other agents are improving their policies in the online learning. Starting from the normalized transition probabilities and changing the transition probabilities following the same optimism principle, the agents increase the values of potential optimal actions optimistically and unanimously, and build consensuses about value estimate. Therefore *transition normalization* and *value deviation* encourage the agents to learn high-performing policies and improve coordination. Moreover, although $\hat{P}_{\mathcal{B}_i}$ is non-stationary (*i.e.*, $\lambda_{vd_i}$ changes over updates of Q-value), we have the following theorem about the convergence of Bellman operator $\mathcal{T}$ under $\hat{P}_{\mathcal{B}_i}$,

$$\mathcal{T}Q_i(s, a_i) = \mathbb{E}_{s' \sim \hat{P}_{\mathcal{B}_i}(s'|s, a_i)}\left[r + \gamma \max_{\hat{a}_i} Q_i\left(s', \hat{a}_i\right)\right].$$

**Theorem 1.** *Under the non-stationary transition probability $\hat{P}_{\mathcal{B}_i}$, the Bellman operator $\mathcal{T}$ is a contraction and converges to a unique fixed point when $\gamma < \frac{r_{\min}}{2r_{\max} - r_{\min}}$, if the reward is bounded by the positive region $[r_{\min}, r_{\max}]$.*

*Proof.* The proof is provided in Appendix A. □

As positive affine transformation of the reward function does not change the optimal policy in the environments with fixed horizon (Zhang et al., 2021), Theorem 1 holds in general. We could rescale the reward to make $r_{\min}$ arbitrarily close to $r_{\max}$ so as to obtain a high upper bound of $\gamma$.

In deep reinforcement learning, directly modifying the transition probability is infeasible. However, we could modify the sampling probability to achieve the same effect. The optimization objective of decentralized deep Q-learning $\mathbb{E}_{p_{\mathcal{B}_i}(s,a_i,s')}|Q_i(s,a_i) - y_i|^2$ is calculated by sampling the batch from $\mathcal{B}_i$ according to the sampling probability $p_{\mathcal{B}_i}(s,a_i,s')$. By factorizing $p_{\mathcal{B}_i}(s,a_i,s')$, we have

$$\underbrace{p_{\mathcal{B}_i}(s,a_i,s')}_{\text{sampling probability}} = p_{\mathcal{B}_i}(s,a_i) * \underbrace{P_{\mathcal{B}_i}(s'|s,a_i)}_{\text{transition probability}}.$$

Therefore, we can modify the transition probability as $\frac{\lambda_{tn_i}\lambda_{vd_i}}{z_i}P_{\mathcal{B}_i}(s'|s,a_i)$ and scale $p_{\mathcal{B}_i}(s,a_i)$ with $z_i$. Then, the sampling probability can be re-written as

$$\underbrace{\lambda_{tn_i}\lambda_{vd_i}p_{\mathcal{B}_i}(s,a_i,s')}_{\text{modified sampling probability}} = z_i p_{\mathcal{B}_i}(s,a_i) * \underbrace{\frac{\lambda_{tn_i}\lambda_{vd_i}}{z_i}P_{\mathcal{B}_i}(s'|s,a_i)}_{\text{modified transition probability}}.$$

Since $z_i$ is independent from $s'$, it could be regarded as a scale factor on $p_{\mathcal{B}_i}(s,a_i)$. Scaling $p_{\mathcal{B}_i}(s,a_i)$ will not change the expected target value $y_i$, so sampling batches for update according to the modified sampling probability could achieve the same effect of modifying the transition probability. Using importance sampling, the modified optimization objective is

$$\mathbb{E}_{\lambda_{tn_i}\lambda_{vd_i}p_{\mathcal{B}_i}(s,a_i,s')}|Q_i(s,a_i) - y_i|^2 = \mathbb{E}_{p_{\mathcal{B}_i}(s,a_i,s')}\frac{\lambda_{tn_i}\lambda_{vd_i}p_{\mathcal{B}_i}(s,a_i,s')}{p_{\mathcal{B}_i}(s,a_i,s')}|Q_i(s,a_i) - y_i|^2$$

$$= \mathbb{E}_{p_{\mathcal{B}_i}(s,a_i,s')}\lambda_{tn_i}\lambda_{vd_i}|Q_i(s,a_i) - y_i|^2,$$

where $\lambda_{tn_i}$ and $\lambda_{vd_i}$ could be seen as the weights of the objective function.

### 3.3 IMPLEMENTATION

We implement our method on BCQ (Fujimoto et al., 2019), termed MABCQ. To make it adapt to high-dimensional continuous spaces, for each agent $i$, we train a Q-network $Q_i$, a perturbation network $\xi_i$, and a conditional VAE $G_i^1 = \{E_i^1\left(\mu^1, \sigma^1|s,a\right), D_i^1\left(a|s,z^1 \sim \left(\mu^1, \sigma^1\right)\right)\}$. In execution, each agent $i$ generates $n$ actions by $G_i^1$, adds small perturbations $\in [-\Phi, \Phi]$ on the actions using $\xi_i$, and then selects the action with the highest value in $Q_i$. The policy can be written as

$$\pi_i(s) = \underset{a_i^j + \xi_i\left(s, a_i^j\right)}{\operatorname{argmax}} Q_i\left(s, a_i^j + \xi\left(s, a_i^j\right)\right), \quad \text{where } \left\{a_i^j \sim G_i^1(s)\right\}_{j=1}^n.$$

$Q_i$ is updated by minimizing

$$\mathbb{E}_{p_{\mathcal{B}_i}(s,a_i,s')}\lambda_{tn_i}\lambda_{vd_i}|Q_i(s,a_i) - y_i|^2, \quad \text{where } y_i = r + \gamma\hat{Q}_i(s', \hat{\pi}_i(s')). \tag{1}$$

$y_i$ is calculated by the target networks $\hat{Q}_i$ and $\hat{\xi}_i$, where $\hat{\pi}_i$ is correspondingly the policy induced by $\hat{Q}_i$ and $\hat{\xi}_i$.

$\xi_i$ is updated by maximizing

$$\mathbb{E}_{p_{\mathcal{B}_i}(s,a_i,s')}\lambda_{tn_i}\lambda_{vd_i}Q_i\left(s, a_i + \xi_i\left(s, a_i\right)\right). \tag{2}$$

To estimate $\lambda_{vd_i}$, we need $V_i^*(s') = \hat{Q}_i(s', \hat{\pi}_i(s'))$ and $\mathbb{E}_{s'}[V_i^*(s')] = \frac{1}{\gamma}(\hat{Q}_i(s,a_i) - r)$, which can be estimated from the sample without actually going through all $s'$. We estimate $\lambda_{vd_i}$ using the target networks to stabilize $\lambda_{vd_i}$ along with the updates of $Q_i$ and $\xi_i$. To avoid extreme values, we clip $\lambda_{vd_i}$ to the region $[1 - \epsilon, 1 + \epsilon]$, where $\epsilon$ is the optimism level.

To estimate $\lambda_{tn_i}$, we train a VAE $G_i^2 = \{E_i^2\left(\mu^2, \sigma^2|s,a,s'\right), D_i^2\left(a|s,s',z^2 \sim \left(\mu^2, \sigma^2\right)\right)\}$. Since the latent variable of VAE follows the Gaussian distribution, we use the mean as the encoding of the input and estimate the probability density functions: $\rho_i(s,a) \approx \rho_{\mathcal{N}(0,1)}(\mu_i^1)$ and $\rho_i(s,a,s') \approx$

---

**Algorithm 1** MABCQ

---

1: **for** $i \in N$ **do**
2:     Initialize the conditional VAEs:
    $G_i^1 = \{E_i^1\left(\mu^1, \sigma^1|s, a\right), D_i^1\left(a|s, z^1\right)\}, G_i^2 = \{E_i^2\left(\mu^2, \sigma^2|s, a, s'\right), D_i^2\left(a|s, s', z^2\right)\}$.
3:     Initialize Q-network $Q_i$, perturbation network $\xi_i$, and their target networks $\hat{Q}_i$ and $\hat{\xi}_i$.
4:     Fit the VAEs $G_i^1$ and $G_i^2$ using $\mathcal{B}_i$.
5:     **for** $t = 1, \ldots, max\_update$ **do**
6:         Sample a mini-batch from $\mathcal{B}_i$.
7:         Update $Q_i$ by minimizing (1).
8:         Update $\xi_i$ by maximizing (2).
9:         Update the target networks $\hat{Q}_i$ and $\hat{\xi}_i$.
10:    **end for**
11: **end for**

---

$\rho_{\mathcal{N}(0,1)}(\mu_i^2)$, where $\rho_{\mathcal{N}(0,1)}$ is the density of unit Gaussian distribution. The conditional density is $\rho_i(s'|a, s) \approx \frac{\rho_{\mathcal{N}(0,1)}(\mu_i^2)}{\rho_{\mathcal{N}(0,1)}(\mu_i^1)}$ and the transition probability is $P_{\mathcal{B}_i}(s'|s, a_i) \approx \int_{s'-\frac{1}{2}\delta_\mathcal{S}}^{s'+\frac{1}{2}\delta_\mathcal{S}} \rho_i(s'|s, a)\mathrm{d}s' \approx \rho_i(s'|s, a)\|\delta_\mathcal{S}\|$ when the integral interval $\|\delta_\mathcal{S}\|$ is a small constant. Approximately, we have

$$\lambda_{tn_i} = \frac{\rho_{\mathcal{N}(0,1)}(\mu_i^1)}{\rho_{\mathcal{N}(0,1)}(\mu_i^2)},$$

and the constant $\|\delta_\mathcal{S}\|$ is considered in $z_i$. In practice, we find that $\lambda_{tn_i}$ falls into the region $[0.2, 1.4]$ for almost all samples. For completeness, we summarize the training procedure of MABCQ in Algorithm 1.

## 4 EXPERIMENTS

### 4.1 MATRIX GAME

We perform MABCQ on the matrix game in Table 1. As shown in Table 3, if we only use $\lambda_{vd}$ without considering *transition normalization*, as the transition probabilities of high-value next states have been increased, for agent 1 the value of $a_2$ becomes higher than that of $a_1$. However, due to the unbalanced action distribution of agent 1, the initial transition probabilities of agent 2 are extremely biased. With $\lambda_{vd}$, agent 2 still underestimates the value of $a_1$ and learns the action $a_2$. The agents arrive at the joint action $(a_2, a_2)$, which is a worse solution than the initial one (Table 2). Normalizing the biased distribution by $\lambda_{tn}$, the agents could learn the optimal solution $(a_2, a_1)$ and build the consensus about the values of learned actions, as shown in Table 4.

Table 3: Transition probabilities and expected returns calculated in the dataset using only $\lambda_{vd}$.

| | action | transition | expected return | | action | transition | expected return |
|---|---|---|---|---|---|---|---|
| Agent 1 | $a_1$ | $p(1|a_1) = 0.12$ 
 $p(5|a_1) = 0.88$ | 4.52 | Agent 2 | $a_1$ | $p(1|a_1) = 0.4$ 
 $p(6|a_1) = 0.6$ | 4 |
| | $a_2$ | $p(6|a_2) = 0.8$ 
 $p(1|a_2) = 0.2$ | 5 | | $a_2$ | $p(5|a_2) = 0.95$ 
 $p(1|a_2) = 0.05$ | 4.8 |

Table 4: Transition probabilities and expected returns calculated in the dataset using $\lambda_{tn}$ and $\lambda_{vd}$.

| | action | transition | expected return | | action | transition | expected return |
|---|---|---|---|---|---|---|---|
| Agent 1 | $a_1$ | $p(1|a_1) = 0.17$ 
 $p(5|a_1) = 0.83$ | 4.33 | Agent 2 | $a_1$ | $p(1|a_1) = 0.14$ 
 $p(6|a_1) = 0.86$ | 5.29 |
| | $a_2$ | $p(6|a_2) = 0.86$ 
 $p(1|a_2) = 0.14$ | 5.29 | | $a_2$ | $p(5|a_2) = 0.83$ 
 $p(1|a_2) = 0.17$ | 4.33 |

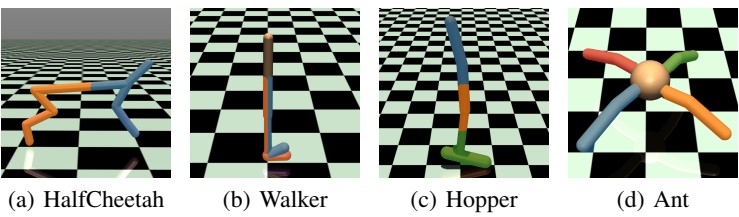

Figure 1: Illustrations of the scenarios. Different colors indicate different agents.

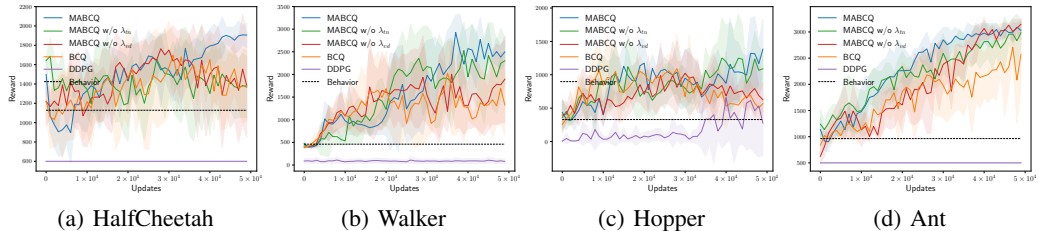

Figure 2: Learning curves of MABCQ and the baselines in HalfCheetah, Walker, Hopper, and Ant. The curves are plotted based the mean and standard deviation of five runs with difference random seeds.

## 4.2 Environments and Datasets

To evaluate the effectiveness of MABCQ in high-dimensional complex environments, we adopt multi-agent mujoco (de Witt et al., 2020), which splits the original action space of the mujoco tasks (Todorov et al., 2012; Brockman et al., 2016) into several sub-spaces. We consider four tasks, which are HalfCheetah, Walker, Hopper, and Ant. As illustrated in Figure 1, different colors indicate different agents. Each agent independently controls one or some joints of the robot and could get the state and reward of the robot, which are defined in the original tasks.

For each environment, we collect $N$ datasets for the $N$ agents. Each dataset contains 1 million transitions $(s, a_i, r, s', done)$. For data collection, we train an intermediate policy and an expert policy for each agent using SAC algorithm (Haarnoja et al., 2018) provided by OpenAI Spinning Up (Achiam, 2018). The offline dataset $\mathcal{B}_i$ is a mixture of four parts: 20% transitions are split from the experiences generated by the SAC agent at the early training, 35% transitions are generated from that the agent $i$ acts the intermediate policy while other agents act the expert policies, 35% transitions are generated from that agent $i$ performs the expert policy while other agents act the intermediate policies, 10% transitions are generated from that all agents perform the expert policies. For the last three parts, we add a small noise to the policies to increase the diversity of the dataset.

We compare MABCQ against the following methods:

- **MABCQ w/o $\lambda_{tn}$**. Removing $\lambda_{tn}$ from MABCQ.
- **MABCQ w/o $\lambda_{vd}$**. Removing $\lambda_{vd}$ from MABCQ.
- **BCQ**. Removing both $\lambda_{tn}$ and $\lambda_{vd}$ from MABCQ.
- **DDPG** (Lillicrap et al., 2016). Each agent $i$ is trained using independent DDPG on the offline $\mathcal{B}_i$ without action constraint and transition probability modification.
- **Behavior**. Each agent $i$ takes the action generated from the VAE $G_i^1$.

The baselines have the same neural network architectures and hyperparameters as MABCQ. All the models are trained for five runs with different random seeds. All the learning curves are plotted using mean and standard deviation. More details about hyperparameters are available in Appendix C.

## 4.3 Performance and Ablation

Figure 2 shows the learning curves of all the methods in the four tasks. Without action constraint and transition probability modification, DDPG severely suffers from the large extrapolation error and can hardly improve the performance throughout the training. BCQ outperforms the behavior policies

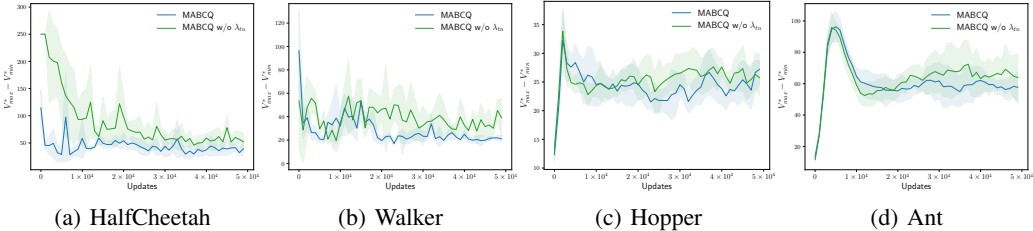

|  (a) HalfCheetah | (b) Walker | (c) Hopper | (d) Ant |

Figure 3: Difference in value estimates among agents along with the training in HalfCheetah, Walker, Hopper, and Ant. It is shown that *transition normalization* indeed reduces the difference in value estimates.

Table 5: Extrapolation errors of MABCQ and BCQ.

|  | HalfCheetah | Walker | Hopper | Ant |
|---|---|---|---|---|
| MABCQ | $98.4 \pm 31.3$ | $\mathbf{55.0} \pm 9.6$ | $\mathbf{28.1} \pm 3.4$ | $\mathbf{180.2} \pm 22.2$ |
| BCQ | $\mathbf{97.2} \pm 29.1$ | $91.5 \pm 35.4$ | $65.8 \pm 6.4$ | $231.3 \pm 47$ |

but only arrives at the mediocre performance. In Figure 2(a) and Figure 2(c), the performance of BCQ even descends in the later stage of learning. During the collection of $\mathcal{B}_i$, when agent $i$ takes a "good" action, other agents usually take "bad" actions, making BCQ underestimate the "good" actions, especially in the latter stage. The learning curves of MABCQ w/o $\lambda_{vd}$ are similar to those of BCQ in the first three tasks. That is because other agents' policies are assumed to be random using only *transition normalization*, which is far from the learned policies and leads to large extrapolation errors. But in Ant, MABCQ w/o $\lambda_{vd}$ outperforms BCQ in the later stage, which is attributed to the value consensus built by the normalized transition probabilities. By optimistically increasing the transition probabilities of high-value next states, MABCQ w/o $\lambda_{tn}$ encourages the agents to learn potential optimal actions and obviously boosts the performance. MABCQ combines the advantages of both value deviation and transition normalization and outperforms other baselines.

To interpret the effectiveness of *transition normalization*, we uniformly sample a subset from the union of all agents' states and calculate the difference in value estimates, $\max_i V_i^* - \min_i V_i^*$, on this subset, where $V_i^*$ is calculated as $Q_i(s, \pi_i(s))$. The results are illustrated in Figure 3. The $\max_i V_i^* - \min_i V_i^*$ of MABCQ is lower than that of MABCQ w/o $\lambda_{tn}$, which verifies that *transition normalization* could decrease the difference in value estimates among agents. If there is a consensus among agents about which states are high-value, the agents would select the actions that most likely lead to the common high-value states. This promotes the coordination of policies and helps MABCQ outperform MABCQ w/o $\lambda_{tn}$.

In Table 5, we present the extrapolation errors of MABCQ and BCQ, $|\frac{1}{N} \sum_i Q_i(s, a_i) - R|$, where $R$ is the true value evaluated by Monte Carlo return. Although MABCQ greatly outperforms BCQ (*i.e.*, higher return), it still achieves smaller extrapolation errors than BCQ in Walker, Hopper, and Ant, which empirically verifies our claim that the proposed weights could decrease the extrapolation error.

We also investigate the effect of optimism level $\epsilon$, the computation efficiency of MABCQ, and the performance of the proposed weights on other datasets and another offline RL algorithm, *i.e.*, CQL (Kumar et al., 2020). Due to the space limit, these results are given in Appendix B.

## 5 CONCLUSION

In this paper, we proposed MABCQ for offline and fully decentralized multi-agent reinforcement learning. MABCQ modifies the transition probability by *value deviation* that increases the transition probabilities of high-value next states, and by *transition normalization* that normalizes the biased transition probabilities. Mathematically, we show that under the non-stationary transition probability after modification, offline decentralized Q-learning converges to a unique fixed point. Empirically, we show that MABCQ could help the agents escape from the suboptimum, learn coordinated policies, and greatly outperform the baselines in a variety of multi-agent offline datasets.

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

## A  PROOFS

**Proposition 1.** *In episodic environments, if each agent $i$ performs Q-learning on $\mathcal{B}_i$, all agents will converge to the same $V^*$ if they have the same transition probability on any state where each agent $i$ acts the learned action $a_i^*$.*

*Proof.* Considering the two-agent case, we define $\delta(s)$ as the difference in the $V^*$.

$$\delta(s) = V_1^*(s) - V_2^*(s)$$
$$= \sum_{s'} P_{\mathcal{B}_1}(s'|s, a_1^*)(r + \gamma V_1^*(s')) - \sum_{s'} P_{\mathcal{B}_2}(s'|s, a_2^*)(r + \gamma V_2^*(s'))$$
$$= \sum_{s'} P_{\mathcal{B}_1}(s'|s, a_1^*)(r + \gamma V_2^*(s') + \gamma V_1^*(s') - \gamma V_2^*(s')) - \sum_{s'} P_{\mathcal{B}_2}(s'|s, a_2^*)(r + \gamma V_2^*(s'))$$
$$= \sum_{s'} (P_{\mathcal{B}_1}(s'|s, a_1^*) - P_{\mathcal{B}_2}(s'|s, a_2^*))(r + \gamma V_2^*(s')) + \gamma P_{\mathcal{B}_1}(s'|s, a_1^*)\delta(s')$$

For the terminal state $s_{end}$, we have $\delta(s_{end}) = 0$. If $P_{\mathcal{B}_1}(s'|s, a_1^*) = P_{\mathcal{B}_2}(s'|s, a_2^*)$, $\forall s' \in S$, recursively expanding the $\delta$ term, we arrive at $\delta(s) = 0 + \gamma 0 + \gamma^2 0 + ... + 0 = 0$. We can easily show that it also holds in the $N$-agent case. $\square$

**Theorem 1.** *Under the non-stationary transition probability $\hat{P}_{\mathcal{B}_i}$, the Bellman operator $\mathcal{T}$ is a contraction and converges to a unique fixed point when $\gamma < \frac{r_{\min}}{2r_{\max}-r_{\min}}$, if the reward is bounded by the positive region $[r_{\min}, r_{\max}]$.*

*Proof.* We initialize the Q-value to be $\eta r_{\min}$, where $\eta$ denotes $\frac{1-\gamma^{T+1}}{1-\gamma}$. Since the reward is bounded by the positive region $[r_{\min}, r_{\max}]$, the Q-value under the operator $\mathcal{T}$ is bounded to $[\eta r_{\min}, \eta r_{\max}]$. Based on the definition of $\hat{P}_{\mathcal{B}_i}(s'|s, a_i)$, it can be written as $\frac{V_i^*(s')}{\sum_{s'} V_i^*(s')}$, where $V_i^*(s') = \max_{\hat{a}_i} Q_i(s', \hat{a}_i)$. Then, we have the following,

$$\|\mathcal{T}Q_i^1 - \mathcal{T}Q_i^2\|_\infty$$

$$= \max_{s,a_i} \left| \sum_{s' \in \mathcal{S}} \hat{P}_{\mathcal{B}_i}^1(s'|s, a_i)\left[r + \gamma \max_{\hat{a}_i} Q_i^1(s', \hat{a}_i)\right] - \sum_{s' \in \mathcal{S}} \hat{P}_{\mathcal{B}_i}^2(s'|s, a_i)\left[r + \gamma \max_{\hat{a}_i} Q_i^2(s', \hat{a}_i)\right] \right|$$

$$= \max_{s,a_i} \gamma \left| \frac{\sum_{s' \in \mathcal{S}}(V_i^{*1}(s'))^2}{\sum_{s' \in \mathcal{S}} V_i^{*1}(s')} - \frac{\sum_{s' \in \mathcal{S}}(V_i^{*2}(s'))^2}{\sum_{s' \in \mathcal{S}} V_i^{*2}(s')} \right|$$

$$= \max_{s,a_i} \gamma \left| \frac{\sum_{s' \in \mathcal{S}}(V_i^{*1}(s'))^2 - (V_i^{*2}(s'))^2}{\sum_{s' \in \mathcal{S}} V_i^{*1}(s')} - \sum_{s' \in \mathcal{S}}(V_i^{*2}(s'))^2\left(\frac{1}{\sum_{s' \in \mathcal{S}} V_i^{*2}(s')} - \frac{1}{\sum_{s' \in \mathcal{S}} V_i^{*1}(s')}\right) \right|$$

$$= \max_{s,a_i} \gamma \left| \frac{\sum_{s' \in \mathcal{S}}(V_i^{*1}(s') - V_i^{*2}(s'))(V_i^{*1}(s') + V_i^{*2}(s'))}{\sum_{s' \in \mathcal{S}} V_i^{*1}(s')} - \sum_{s' \in \mathcal{S}}(V_i^{*2}(s'))^2 \frac{\sum_{s' \in \mathcal{S}} V_i^{*1}(s') - V_i^{*2}(s')}{\sum_{s'} V_i^{*1}(s') \sum_{s' \in \mathcal{S}} V_i^{*2}(s')} \right|$$

$$\leq \max_{s,a_i} \gamma \left| \sum_{s' \in \mathcal{S}}(V_i^{*1}(s') - V_i^{*2}(s')) \right| * \frac{1}{\sum_{s' \in \mathcal{S}} V_i^{*1}(s')} * \max \left| (V_i^{*1}(s') + V_i^{*2}(s')) - \frac{\sum_{s' \in \mathcal{S}}(V_i^{*2}(s'))^2}{\sum_{s' \in \mathcal{S}} V_i^{*2}(s')} \right|$$

$$\leq \gamma|\mathcal{S}|\|Q_i^1 - Q_i^2\|_\infty * \frac{1}{|\mathcal{S}|\eta r_{\min}} * \eta(2r_{\max} - r_{\min})$$

$$= \gamma\left(\frac{2r_{\max}}{r_{\min}} - 1\right)\|Q_i^1 - Q_i^2\|_\infty.$$

The third term of the penultimate line is because: if $V_i^{*1}(s') + V_i^{*2}(s') > \frac{\sum_{s' \in \mathcal{S}}(V_i^{*2}(s'))^2}{\sum_{s' \in \mathcal{S}} V_i^{*2}(s')}$,

$$V_i^{*1}(s') + V_i^{*2}(s') - \frac{\sum_{s' \in \mathcal{S}}(V_i^{*2}(s'))^2}{\sum_{s' \in \mathcal{S}} V_i^{*2}(s')} \leq V_i^{*1}(s') + V_i^{*2}(s') - \frac{\sum_{s' \in \mathcal{S}}(V_i^{*2}(s')) * \eta r_{\min}}{\sum_{s' \in \mathcal{S}} V_i^{*2}(s')} \leq 2\eta r_{\max} - \eta r_{\min},$$

else,

$$\frac{\sum_{s'\in\mathcal{S}}(V_i^{*^2}(s'))^2}{\sum_{s'\in\mathcal{S}}V_i^{*^2}(s')} - (V_i^{*^1}(s') + V_i^{*^2}(s')) \leq \frac{\sum_{s'\in\mathcal{S}}(V_i^{*^2}(s'))*\eta r_{\max}}{\sum_{s'\in\mathcal{S}}V_i^{*^2}(s')} \leq \eta r_{\max}.$$

Since $2\eta r_{\max} - \eta r_{\min} \geq \eta r_{\max}$, we have

$$|(V_i^{*^1}(s') + V_i^{*^2}(s')) - \frac{\sum_{s'\in\mathcal{S}}(V_i^{*^2}(s'))^2}{\sum_{s'\in\mathcal{S}}V_i^{*^2}(s')}| \leq 2\eta r_{\max} - \eta r_{\min}.$$

Therefore, if $\gamma < \frac{r_{\min}}{2r_{\max}-r_{\min}}$, the operator $\mathcal{T}$ is a contraction. By contraction mapping theorem, $\mathcal{T}$ converges to a unique fixed point. $\qquad\square$

## B   ADDITIONAL RESULTS

To clearly present the performance gain of MABCQ over BCQ, we summarize the mean value and standard deviation after $5 \times 10^4$ updates in Table 6, where we can see that MABCQ performs more than one standard deviation better than BCQ.

Table 6: Comparison between MABCQ and BCQ.

|  | HalfCheetah | Walker | Hopper | Ant |
|---|---|---|---|---|
| MABCQ | $\mathbf{1905 \pm 126}$ | $\mathbf{2498 \pm 256}$ | $\mathbf{1383 \pm 443}$ | $\mathbf{3027 \pm 178}$ |
| BCQ | $1384 \pm 360$ | $1324 \pm 662$ | $566 \pm 89$ | $2573 \pm 637$ |

To evaluate MABCQ in low-quality dataset, we carried out an additional experiment on 2-agent Swimmer with another data collection approach, where each dataset only contains the experiences at the early training and the experiences generated by intermediate policies. As shown in Figure 4, after removing the expert experiences, MABCQ greatly outperforms BCQ, and BCQ even cannot outperform the behavior policies.

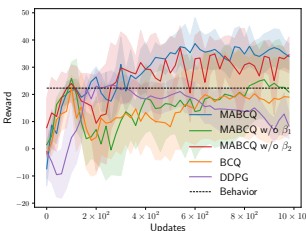

Figure 4: Learning curves of MABCQ and the baselines in Swimmer.

The optimism level $\epsilon$ controls the strength of *value deviation*. If $\epsilon$ is too small, *value deviation* has weak effects on the objective function. But if $\epsilon$ is too large, the agent will be over optimistic about other agents' learned policies, which could result in large extrapolation errors and uncoordinated policies. Figure 5 shows the learning curves of MABCQ with different $\epsilon$. It is commonly observed that increasing $\epsilon$ elevates the performance, especially in HalfCheetah. However, in Walker and Ant,

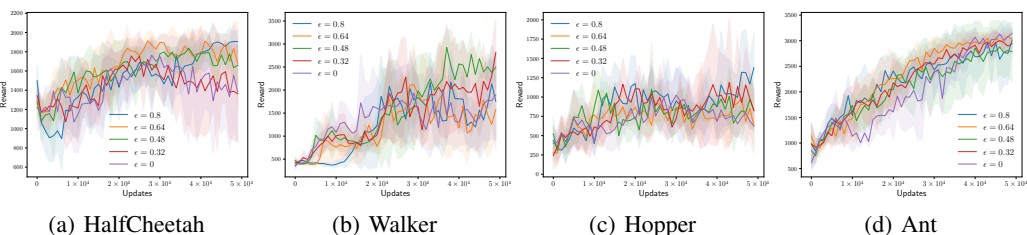

(a) HalfCheetah       (b) Walker       (c) Hopper       (d) Ant

Figure 5: Learning curves of MABCQ with different $\epsilon$ in HalfCheetah, Walker, Hopper, and Ant. It is shown that with any positive $\epsilon$, MABCQ does not underperform that without *value deviation*.

Table 7: Average time taken by one update.

|        | HalfCheetah | Walker | Hopper | Ant   |
|--------|-------------|--------|--------|-------|
| MABCQ  | 18 ms       | 18 ms  | 16 ms  | 20 ms |
| BCQ    | 10 ms       | 10 ms  | 9 ms   | 11 ms |

if we set a large $\epsilon$ (0.8), the performance slightly drops due to overoptimism. Even so, with any positive $\epsilon$, MABCQ does not underperform that without *value deviation*.

To demonstrate the computation efficiency of our method, we record the average time taken by one update in Table 7. The experiments are carried out on Intel i7-8700 CPU and NVIDIA GTX 1080Ti GPU. Since $\lambda_{vd}$ and $\lambda_{tn}$ could be calculated from the sampled experience without actually going through all next states, MABCQ additionally needs only two forward passes for computing $\lambda_{vd}$ and $\lambda_{tn}$ in the update and costs less than twice the training time of BCQ. Moreover, since MABCQ is fully decentralized, the learning of agents can be entirely parallelized. Therefore, MABCQ scales well with the number of agents.

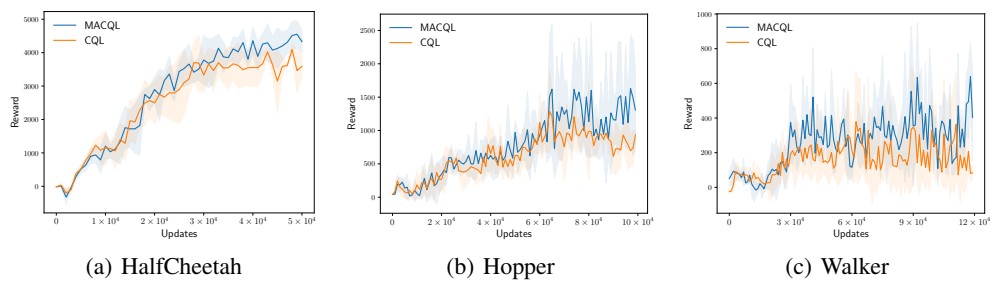

| (a) HalfCheetah | (b) Hopper | (c) Walker |
|-----------------|------------|------------|

Figure 6: Learning curves of MACQL in D4RL medium-replay datasets.

The two proposed weights $\lambda_{vd}$ and $\lambda_{tn}$ could also be extended to other offline RL methods. We introduce the two weights to CQL (Kumar et al., 2020), as MACQL, and test them in D4RL medium-replay datasets (Fu et al., 2020). CQL augments the standard Bellman error objective with conservative regularizer, thus the learned Q-value may not represent the original meaning and thus not theoretically match value deviation. However, MACQL still outperforms CQL, as shown in Figure 6.

## C EXPERIMENTAL SETTINGS AND HYPERPARAMETERS

The experimental settings and hyperparameters are summarized in Table 8.

Table 8: Experimental settings and hyperparameters

| Hyperparameter         | HalfCheetah | Walker    | Hopper | Ant  |
|------------------------|-------------|-----------|--------|------|
| agent number ($N$)     | 2           | 2         | 3      | 4    |
| state space            | 17          | 17        | 11     | 27   |
| action space           | 3           | 3         | 1      | 2    |
| horizon ($T$)          |             | 1000      |        |      |
| discount ($\gamma$)    |             | 0.99      |        |      |
| $\mathcal{B}_i$ size   |             | $10^6$    |        |      |
| batch size             |             | 1024      |        |      |
| MLP units              |             | $(64, 64)$|        |      |
| MLP activation         |             | ReLU      |        |      |
| learning rate of $Q$   |             | $10^{-3}$ |        |      |
| learning rate of $\xi$ |             | $10^{-4}$ |        |      |
| learning rate of $G$   |             | $10^{-4}$ |        |      |
| $\epsilon$             | 0.80        | 0.48      | 0.80   | 0.64 |
| $\Phi$                 |             | 0.05      |        |      |
| $n$                    |             | 10        |        |      |
| VAE hidden space       |             | 10        |        |      |

## D    LIMITATION AND FUTURE WORK

Although we consider the setting where each agent can get the state of the environment, MABCQ could also be potentially applied to partially observable environments. However, if the partial observability is too limited, Q-value may not be accurately estimated from the observation. Many MARL methods adopt recurrent neural networks to utilize the historical information, which however is impractical if the timestamp is not included in the offline dataset. Moreover, if a state is estimated as a high-value state by an agent but not included in the datasets of other agents, the other agents cannot learn corresponding optimal actions to cooperate with that agent at that state, and thus miscoordination may occur. In this case, each agent should conservatively estimate the values of states that are absent from the datasets of other agents to avoid miscoordination. We leave the observation limitation and the state absence to future work.

