# OpenReview forum: "Offline Decentralized Multi-Agent Reinforcement Learning"
_ICLR.cc/2022/Conference — ICLR 2022 Submitted_

### Official Review · Reviewer_WD4d · 2021-10-31

**Correctness:** 2
**Technical Novelty And Significance:** 2
**Empirical Novelty And Significance:** 2
**Recommendation:** 3
**Confidence:** 5

**Main Review:**

Originality: The problem the paper studies is important and interesting.

Quality:
1.Value deviation aims to increase the transition probabilities of the high-value next states, but it will increase the value estimate of the current state and action. It can worsen the performance in bad datasets, which is discussed in the CQL paper.
2.The evaluation of MABCQ is not sufficient, the dataset seems to be "replay" in RL domain. How does  MABCQ perform in other datasets: Random, Medium, Expert?
3.The interpretation of the effectiveness of transition normalization in Figure 3 is not solid. The difference between the maximum value and minimum value can not quantify the coordination of agents, it would be better to show a real example.

Soundness:
1. A recent MARL offline RL paper "Believe What You See: Implicit Constraint Approach
for Offline Multi-Agent Reinforcement Learning" is not discussed and compared.
2. The weakness of the proposed heuristic method is not discussed, is there any other method that can improve the coordination of agents?


**Summary Of The Paper:**

This paper studies a MARL offline setting, where each agent is trained independently. value deviation and transition normalization techniques are proposed to modify the transition probabilities. Experimental results on four benchmarks show MABCQ's performance improvement over BCQ.

**Summary Of The Review:**

This paper studies a novel problem in MARL area. However, the proposed value deviation method is not well motivated, the experimental evaluation is only done in replay datasets, and other marl offline paper are not cited and discussed. Thus I recommend the rejection.

---

> ### Author Response · Authors · 2021-11-16
> **Response to Reviewer WD4d**
>
> > Value deviation aims to increase the transition probabilities of the high-value next states, but it will increase the value estimate of the current state and action. Value deviation is not well motivated.
>
> The "overestimate" caused by value deviation is exactly what we need. For example, in offline datasets of the matrix game (Table 1), there is no out-of-distribution action, but the agents still converge to the local optimal (Table 2). Why? Because the behavior policy of agent 1 is very poor, if agent 2 takes the optimal policy $a_1$, the probability of transferring to the high-value state $6$ is extremely low, therefore, agent 2 underestimates the value of  $a_1$ and learns the suboptimal action $a_2$. Since the behavior policy of agent 1 is very poor, it is a bad dataset. MABCQ does not worsen the performance in this bad dataset but makes the agents learn optimal actions (Table 4). To avoid underestimation, value deviation lets each agent be optimistic toward other agents and increases the Q values. The "overestimate" caused by value deviation could decrease the extrapolation errors and help the agents escape from the suboptimum.
>
> Value deviation is motivated by other optimistic decentralized MARL methods. For example,  Lenient-DQN[1] introduces optimism in the value function update by forgiving suboptimal actions.
>
> Moreover, Since MABCQ is instantiated on BCQ, it could avoid the overestimation caused by out-of-distribution actions.
>
> >  How does MABCQ perform on other datasets?
>
> Since MABCQ addresses the problem of transition bias, it is suitable for the datasets that are independently collected by multi-modal policies, e.g., the datasets in Section 4.2. The results of MABCQ in D4RL are summarized in Table 3. In the datasets we built, the datasets of agents are independently collected. But in D4RL datasets, the datasets of agents are jointly collected, where the behavior policies $j$ in dataset $i$ and dataset $j$ are the same, and trained by BCQ, the learned policy $j$ is colse to behavior policy $j$, thus the offline transition dynamics of agent $i$ are colse to the online transition dynamics. Therefore, the performance gain of MABCQ is not obvious in D4RL random and medium datasets.
>
> Table 3: The performance of MABCQ in D4RL.
>
> |                    | MABCQ         | BCQ           |
> | ------------------ | ------------- | ------------- |
> | halfcheetah-random | $1086 \pm 12$ | $989 \pm 14$  |
> | walker-random      | $317 \pm 4$   | $310\pm 14$   |
> | halfcheetah-medium | $3677 \pm 94$ | $3536 \pm 45$ |
> | walker-medium      | $863\pm 32$​   | $780\pm 22$   |
>
> In Appendix, we evaluate MABCQ in the low-quality dataset, which only contains the experiences at the early training and the experiences generated by intermediate policies, without expert experiences. Also, we introduce the two proposed weights to CQL as MACQL, and test them in D4RL medium-replay datasets.  MACQL outperforms CQL, which shows the two weights could obtain performance gain on other types of datasets and other offline RL methods.
>
> > The difference between the maximum value and minimum value can not quantify the coordination of agents.
>
> The aim of plotting Figure 3 is not to quantify the coordination of agents but to verify our claim that transition normalization reduces the difference in value estimates. The coordination should be quantified by mean reward, which is shown in Figure 2, where MABCQ outperforms MABCQ w/o $\lambda_{tn}$.
>
> > MA-ICQ (Yang et al., 2021)
>
> MA-ICQ is a centralized-training method that requires joint actions, but we adopt settings of fully decentralized training, where the joint actions is not available. So MA-ICQ cannot be applied to decentralized datasets. We will include the discussion about MA-ICQ in the revision.
>
> > The weakness of the proposed heuristic method is not discussed. Is there any other method that can improve the coordination of agents?
>
> We have discussed the limitations of MABCQ in Appendix. To the best of our knowledge, MABCQ is the first method to improve the coordination in offline and decentralized settings.
>
> [1]https://arxiv.org/pdf/1707.04402.pdf

---

### Official Review · Reviewer_zmVr · 2021-11-01

**Correctness:** 3
**Technical Novelty And Significance:** 2
**Empirical Novelty And Significance:** 2
**Recommendation:** 3
**Confidence:** 3

**Main Review:**

I appreciate the ideas proposed by the authors. However, a lot of descriptions in the submission are very vague and the statements are mathematically imprecise. I also have some concerns regarding the generality of the proposed framework. Please find my detailed comments below.

(1) A lot of sentences are very vague and imprecise (without introducing the framework in a rigorous) way. To name a few examples:

(1-A) It's stated in the abstract that "However, the transition probabilities calculated from the dataset can be much different from the transition probabilities induced by the learned policies of other agents..." This really depends on the information observability of the agents. If an individual agent could observe the states and actions of all agents in the system, then this won't be an issue.

(1-B) In the first paragraph, the authors keep switching between "agents" and "agent" and sometimes mentioned "offline RL", it is difficult to understand if you are always talking about the multi-agent setting or moving to talk about the single-agent setting at some point.

(1-C) The authors start with "The main challenge of offline RL is the extrapolation error" in the second paragraph, are you referring to the single-agent case, or multi-agent case, or both? Later in the paragraph, the authors mentioned "... ignore to correct the transition bias" without proper explanation of what you mean by transition bias.

(1-D) The authors mentioned multiple times on "decentralized multi-agent environment" which is confusing to me. Based on my understanding of the paper, the proposed learning procedure is decentralized (choice of the agent or the coordinator) but this has nothing to do with the environment.

(1-E) The second paragraph on page 2 does not make sense to me before talking about what is observable and what is not abservable to each individual agent.

(1-F) I don't understand the sentence "If the transition probabilities of high-value next states are extremely low..."

(2) Set up in Section 3.1:

(2-A) The problem description is not mathematically rigorous. The authors should specify the dimension or the space.

(2-B) Why the reward only depends on the state (not the action)?

(2-C) It reads to me that the centralized training and decentralized execution scheme could still be applied to this setting. In this case, the bias issue mentioned in the motivation should be solved.




**Summary Of The Paper:**

This paper proposes a method named MABCQ to utilize offline data in the MARL environment via (1) value deviation and (2) transition normalization. These two techniques are used to correct the bias in the individual observation of the transition probability and improve the discovery of optimal policy. The authors showed the convergence of the Q-learning under the non-stationary transition probabilities after modification.



**Summary Of The Review:**

A lot of descriptions in the submission are very vague and the statements are mathematically imprecise. I do not think the submission is ready for acceptance with its current presentation.

---

> ### Author Response · Authors · 2021-11-16
> **Response to Reviewer zmVr**
>
> 1-A. We have clearly formulated the offline and decentralized settings in the first paragraph and Section 3.1, where the agent cannot observe the actions of other agents.
>
> 1-B. Since we consider the decentralized settings, we have to illustrate the motivation and formulation from the perspective of the individual agent, that is why we use "agent".
>
> 1-C. Obviously, extrapolation error is the main challenge in both single-agent and multi-agent offline RL. In the second paragraph, we discuss the two sources of extrapolation error: the distance of the learned action distribution to the behavior action distribution, and the bias of the transition dynamics estimated from the dataset to the true transition dynamics. The second one is simplified as transition bias.
>
> 1-D. Decentralized learning is very related to the environment. If communication is not allowed in the environment, the agent cannot obtain the information of other agents and could only learn in a decentralized way.
>
> 1-E. We have clearly stated that each individual agent cannot obtain the actions of other agents in the first paragraph on page 1.
>
> 1-F. For example, in the matrix game (Table 1), since the behavior policy of agent 1 is very poor, if agent 2 takes the optimal policy $a_1$, the probability of transferring to the high-value state $6$ is extremely low, therefore, agent 2 underestimates the value of  $a_1$ and learns the suboptimal action $a_2$ (Table 2).
>
> 2-A. In fact, we have specified the state space and the action space.
>
> 2-B. The motivation of this paper is to investigate the extrapolation error caused by the bias in transition probabilities, so we simplify the reward function to a dependence on the current state, which is a common assumption in reinforcement learning, e.g., in TRPO and PPO.
>
> 2-C. Centralized training cannot be applied in this setting, since each individual agent cannot obtain the actions of other agents.  Also, the dataset of each agent is collected independently, which means the state sets of the agents are different.

---

### Official Review · Reviewer_N2SH · 2021-11-03

**Correctness:** 3
**Technical Novelty And Significance:** 3
**Empirical Novelty And Significance:** 2
**Recommendation:** 3
**Confidence:** 2

**Main Review:**

It is not clear to me when the setting described by the authors occurs in practical cases. I think that the paper lacks a clear applicative example in which the framework proposed by the authors occurs.

The assumption that the transition probabilities are deterministic for the environment seems to be a bit restrictive. Again, if the author could provide a practical example, it would help to motivate the setting described by the authors.

In the first formula of page 5, the first two terms cancel out. I suggest to provide a more direct formula for this probability, e.g., P_B_i(s'|s,a_i) = 1/|S|. Moreover, the presentation of these modifications of the probabilities should be more formally defined with a proper symbol for each one of them.

Proposition 1 is misleading. If I understood correctly the theorem holds if one applies Q-learning over the previously modified transitions. It should be mentioned in the theorem statement.

The title of Section 3.3 (Implementation) is misleading. First, I would have mentioned which are the characteristics and the assumptions required for a generic methodology to be compliant with your proposed approach. Second, this seems a specific case of the application of your method rather than its implementation.

I think that the experiments, even if they present some difference in the expected values of the agents' objective, do not provide any statistical significance of the fact that the proposed method is somehow providing an improvement over the existing ones (other than DDPG).

It would also be interesting to see the performance of fully centralized learning, to understand how much we are losing from the fact that we are learning in a decentralised manner.

**Summary Of The Paper:**

The authors propose a scheme to modify the transition probabilities in the learning process in a fully decentralised MARL setting. The authors show that the method is still converging to the optimal solution. They also provide some experiments on synthetic environments.

**Summary Of The Review:**

The topic is interesting, but the setting lacks a strong applicative example and the formalism of the paper should be improved before it is worthed for publication. Moreover, I think that the statements sometimes are hard to understand. Therefore, I suggest the author should try their best to improve the readability of the paper. Moreover, the experiments should provide some statistical significance that the proposed method is better than the state of the art.

---

> ### Author Response · Authors · 2021-11-16
> **Response to Reviewer N2SH**
>
> > An applicative example of offline and decentralized settings.
>
> The setting of offline and decentralized training could apply to many decentralized industrial systems, where autonomous driving is the most important one, as we have pointed out in Introduction. First, each car cannot obtain the actions of other cars, thus centralized training is impracticable. Second, in autonomous driving, consistent online interaction is costly and risky, so we have to train agents from offline experiences collected by human drivers.
>
> > Applicative examples of deterministic settings.
>
> For the popular multi-agent simulations, i.e., MPE, SMAC, and multi-agent mujoco, the environment is deterministic. For practical examples, if we consider all of the cars in autonomous driving, the environment is deterministic.
>
> > The first formula of page 5.
>
> The reason for writing $P_{B_{i}}\left(s^{\prime} \mid s, a_{i}\right) * \frac{1}{P_{B_{i}}\left(s^{\prime} \mid s, a_{i}\right)} * \frac{1}{z_{i}^{t n}}$​ is to clearly show that we need to estimate $\frac{1}{P_{B_{i}}\left(s^{\prime} \mid s, a_{i}\right)}$​ as $\lambda_{tn}$​.
>
> > Proposition 1
>
> Proposition 1 holds if the agents have the same transition probability on any state and learned action. Modified by transition normalization, the transition probabilities meet the requirement.
>
> > Significance of experiments.
>
> In Appendix, we summarize the mean value and standard deviation after $5 \times 10^4$ updates in Table 6, where we can see that MABCQ performs more than one standard deviation better than BCQ.
>
> > It would also be interesting to see the performance of fully centralized learning, to understand how much we are losing from the fact that we are learning in a decentralised manner.
>
> Centralized learning cannot be performed on decentralized datasets that we built, since the dataset of each agent is collected independently, which means the state sets of the agents are different. A state $s_t$ in the dataset of agent $i$​​​​​​ can be *not* included in the datasets of other agents.
>
> In Appendix, we have reported the performance of decentralized CQL and MACQL (adding the proposed weights to CQL[1] in D4RL[2] medium-replay datasets. In such datasets, agents have the same state sets. The performance centralized training is reported in the D4RL paper, and we summarize it in Table 2. We observe the performance drop in decentralized learning.
>
> Table 2: The performance of centralized training and decentralized training.
>
> |             | Centralized | Decentralized |
> | ----------- | ----------- | ------------- |
> | HalfCheetah | 5455.6      | 3589.7        |
> | Hopper      | 1227.3      | 933.1         |
> | Walker      | 1227.3      | 82.7          |
>
> [1]https://arxiv.org/pdf/2006.04779.pdf
>
> [2]https://arxiv.org/pdf/2004.07219.pdf

---

### Official Review · Reviewer_Hcfz · 2021-11-04

**Correctness:** 2
**Technical Novelty And Significance:** 2
**Empirical Novelty And Significance:** Not applicable
**Recommendation:** 5
**Confidence:** 4

**Main Review:**

Pros:
1. The problem studied in this paper is very interesting, with a clear motivating example emphasizing that in the decentralized offline MARL setting, the difference between transition probabilities in the dataset and other agents can be large and harm the performance. Overall, the paper is well-written.
2. The theoretical analysis shows that the proposed operator is able to converge.

Cons:
1. There is a recent paper about offline MARL, MA-ICQ (Yang et al., 2021) which should be compared or at least discussed in the paper.
2. The assumption in Theorem that gamma<r_min/(2r_max-r_min) seems too strong in the case that r_max>>r_min.
3. The experimental evaluations are not sufficient as it only compares MABCQ with baselines on a mixed dataset. How does MABCQ perform on other types of datasets?



**Summary Of The Paper:**

The paper studies an offline MARL setting, with a focus on the decentralized case, and discovers that the difference between transitions in the dataset and other agents can be very large. Authors propose two techniques, value deviation and transition normalization, to modify the transition probabilities. Experimental results validate its performance improvement over BCQ on multi-agent MuJoCo tasks.


**Summary Of The Review:**

This paper studies an interesting offline decentralized MARL setting. However, relevant works are not cited and discussed, and experimental evaluation can be improved.

---

> ### Author Response · Authors · 2021-11-16
> **Response to Reviewer Hcfz**
>
> > MA-ICQ (Yang et al., 2021)
>
> MA-ICQ is a centralized-training method that requires joint actions, but we adopt settings of fully decentralized training, where the joint actions are not available. So MA-ICQ cannot be applied to decentralized datasets. We will include the discussion about MA-ICQ in the revision.
>
> > The assumption in Theorem that gamma<r_min/(2r_max-r_min) seems too strong
>
> In Section 3.2.3, we have pointed out that positive affine transformation of the reward function does not change the optimal policy in the environments with the fixed horizon. Therefore Theorem 1 holds in general, and we could rescale the reward to make r_min arbitrarily close to r_max so as to obtain a high upper bound of $\gamma$. When r_min is close to r_max, $\gamma$ will be close to $1$.
>
> >  How does MABCQ perform on other types of datasets?
>
> Since MABCQ addresses the problem of transition bias, it is suitable for the datasets that are independently collected by multi-modal policies, e.g., the datasets in Section 4.2. In Appendix, we evaluate MABCQ in the low-quality dataset, which only contains the experiences at the early training and the experiences generated by intermediate policies, without expert experiences. We also introduce the two proposed weights to CQL as MACQL, and test them in D4RL medium-replay datasets.  MACQL outperforms CQL, which shows the two weights could obtain performance gain on other types of datasets and other offline RL methods.
>
> The results of MABCQ in D4RL are summarized in Table 1. In the datasets we built, the datasets of agents are independently collected. But in D4RL datasets, the datasets of agents are jointly collected, where the behavior policies $j$ in dataset $i$ and dataset $j$ are the same, and trained by BCQ, the learned policy $j$ is colse to behavior policy $j$, thus the offline transition dynamics of agent $i$ are colse to the online transition dynamics. Therefore, the performance gain of MABCQ is not obvious in D4RL random and medium datasets.
>
> Table 1: The performance of MABCQ in D4RL.
>
> |                    | MABCQ         | BCQ           |
> | ------------------ | ------------- | ------------- |
> | halfcheetah-random | $1086 \pm 12$ | $989 \pm 14$  |
> | walker-random      | $317 \pm 4$   | $310\pm 14$   |
> | halfcheetah-medium | $3677 \pm 94$ | $3536 \pm 45$ |
> | walker-medium      | $863\pm 32$​  | $780\pm 22$   |

---

### Decision · Program_Chairs · 2022-01-20

**Decision:**

Reject

**Comment:**

This paper studies the offline multi-agent RL problem. The finding is that the dataset collected by one agent could be very different for other agents. The authors provide two solutions to this problem. Although being interesting, the reviewers found that the there are many imprecise math statements, and some of the methods are not well motivated. Hence, the overall recommendation is a reject.